# Study of Micro/Nano Structuring and Mechanical Properties of KrF Excimer Laser Irradiated Al for Aerospace Industry and Surface Engineering Applications

**DOI:** 10.3390/ma14133671

**Published:** 2021-06-30

**Authors:** Nisar Ali, Shazia Bashir, Ali Mohammad Alshehri, Narjis Begum

**Affiliations:** 1Department of Basic Sciences and Humanities, University of Engineering and Technology Lahore, Kala shah Kaku Campus, GT Road, Sheikhupura 39020, Pakistan; ummikalsoom14@gmail.com; 2Laser Laboratories, Centre for Advanced Studies in Physics, Government College University, 1-Church Road, Lahore 54000, Pakistan; shaziabashir@gcu.edu.pk; 3Department of Basic Sciences and Humanities, University of Engineering and Technology Lahore, Faisalabad Campus, Faisalabad 37630, Pakistan; 4Department of Physics, King Khalid University, P.O. Box 9004, Abha 61413, Saudi Arabia; amshehri@kku.edu.sa; 5Department of Physics, Comsats Institute of Information Technology, Islamabad 12000, Pakistan; narjisgh@hotmail.com

**Keywords:** laser ablation, aluminum, ambient environment, correlation, surface structuring, crystallinity, alumina, nanohardness

## Abstract

Micro/nano structuring of KrF Excimer laser-irradiated Aluminum (Al) has been correlated with laser-produced structural and mechanical changes. The effect of non-reactive Argon (Ar) and reactive Oxygen (O_2_) environments on the surface, structural and mechanical characteristics of nano-second pulsed laser-ablated Aluminum (Al) has been revealed. KrF Excimer laser with pulse duration 20 ns, central wavelength of 248 nm and repetition rate of was utilized for this purpose. Exposure of targets has been carried out for 0.86, 1, 1.13 and 1.27 J^·^cm^−2^ laser fluences in non-reactive (Ar) and reactive (O_2_) ambient environments at a pressure of 100 torr. A variety of characteristics of the irradiated targets like the morphology of the surface, chemical composition, crystallinity and nano hardness were investigated by using Scanning Electron Microscopy (SEM), Atomic Force Microscopy (AFM), Energy Dispersive X-ray Spectroscopy (EDS), X-ray Diffractometer (XRD), Raman spectroscopy and Nanohardness tester techniques, respectively. The nature (reactive or non-reactive) and pressure of gas played an important role in modification of materials. In this study, a strong correlation is observed between the surface structuring, chemical composition, residual stress variation and the variation in hardness of Al surface after ablation in both ambient (Ar, O_2_). In the case of reactive environment (O_2_), the interplay among the deposition of laser energy and species of plasma of ambient gas enhances chemical reactivity, which causes the formation of oxides of aluminum (AlO, Al_2_O_3_) with high mechanical strength. That makes it useful in the field of process and aerospace industry as well as in surface engineering.

## 1. Introduction

Pulsed laser ablation of solids is a promising technique and has variety of uses in laser-induced breakdown spectroscopy, surface structuring [1], manufacturing, micromachining, microelectronics and thin film deposition [2,3,4,5].

Laser induced structuring develops various kinds of structures like Laser Induced Periodic Surface Structures (LIPSS), grids, dots and many complex structures after being inspired from nature like from butterfly wings, shark skin, lotus leaf, etc. [6]. Hierarchical structures can also be developed using lasers with many important properties, such as super hydrophilic/super hydrophobic effects, self-cleaning and anti-icing properties on laser-irradiated metals [6,7,8]. Surface structuring has tremendous technological applications, such as in the field of industry [9], biomedical [10] and optical sciences by growing various coatings and micro/nanostructures that are efficient for light detection, reflection, data storage, field emission and desired tribological properties of surfaces [11,12]. Laser-induced structuring can also cause the enhancement in surface roughness and increased adhesion of irradiated target surface [13]. During nanosecond laser material interactions, the thermal processes become dominant and cause the variation in surface roughness [6,14].

The reactive or non-reactive nature and pressure of ambient gas are the chief controlling factors for the characteristics of plasma and play a momentous role for the modification of the material [15]. Shielding effect, spatial confinement and slowing down of the expanding plasma plume are the common effects of the background gas [15]. The interplay among deposition of laser energy and the species of plasma of the ambient gas (O_2_, H_2_, N_2_) during irradiation of the material enhances the chemical reactivity, which causes the development of oxides, hydrides and nitrides [16] on the ablated target surface.

The composition and surface structure of laser-ablated Al after ablation in reactive (O_2_ from air) and non-reactive (Ar) environments has been reported by Balchev et al. [17]. Maximum redeposited oxides are observed in the case of air treatment while small structures like nano particles are created in Ar ambient. Ali et al. [18] studied the development of various kind of features after laser irradiation of metallic targets in air and liquid environments. They observed the enhancement in surface hardness after irradiation in air ambient as compared to liquid ambient.

In the current research work, Al is chosen as target due to large range of its applications in electronics, space science and industry [19]. It is frequently utilized owing to its low density, extraordinary specific strength, high thermal conductivity, ductility and good corrosion resistance at room temperature. On the other hand, its poor wear resistance and hardness limits its usages in wearing and harsh environments [20]. These shortcomings can be overcome by strengthening the Al target’s surface through laser-induced structuring in non-reactive environment and through laser-induced structuring and the synthesis of Al_2_O_3_ in reactive environment (O_2_) [21,22].

Interaction between high power laser beam and target implicates high rates of heating/cooling and temperature gradients, generating metastable phase, causing a wide range of surface micro-structures with innovative properties that cannot be developed using conventional processing tools [23,24]. This may lead to improve corrosion, mechanical and tribological properties of irradiated target materials [19]. The properties of the surface can also be enhanced by oxidation of the surface of metal along with the creation of surface structures. Pulsed laser ablation of target surface in O_2_ environment is well known method for oxidation of the target surface to produce oxides of Al (AlO, Al_2_O_3_) [25,26,27]. The aim behind the modification of surface and structural properties of Al after pulsed laser ablation in Ar and O_2_ is to modify broad range functional as well as physical and chemical characteristics. The ablation of Al in O_2_ ambient results in the formation of Al_2_O_3_ (alumina). Alumina is a high-performance ceramic material with excellent temperature resistance, high mechanical strength and good tribological properties. All these properties make it useful in the field of process and aerospace industry as well as in surface engineering [28]

The aims of the present work are (a) to explore the consequence of laser fluences and gaseous environment on surface structuring, chemical composition and mechanical characteristics of Al and to synthesize alumina, and (b) to correlate the surface structuring with the structural and mechanical changes produced after irradiation with nanosecond laser. The morphological variations of irradiated targets were studied by Scanning Electron Microscope (SEM). The chemical composition and phases are measured for unirradiated and laser-ablated targets by using Energy Dispersive X-ray Spectroscopy (EDS), X-ray Diffractometer (XRD) and Raman analysis. Nanohardness measurement analysis is utilized to study the variation in surface hardness.

In this study, a strong correlation is observed between the surface structuring, chemical composition, residual stresses and variation in surface hardness of Al after ablation in both ambient (Ar, O_2_). One other novelty of the current research work is that no or rare work is performed on the correlation of surface structures with structural as well as mechanical properties and synthesis of alumina (Al_2_O_3_) after ablation of Al in the above-mentioned environments using KrF Excimer lasers at the selected parameters. Mostly, the literature describes ablation of Al targets under vacuum and air.

O_2_-aided exothermic reactions become dominant during laser irradiation of metals in O_2_ ambient, which causes reduction in the ablation threshold of the material and makes laser material processing useful for welding, cutting, drilling and in the field of industry.

## 2. Experimental Setup

KrF Excimer laser (EX GAM ORLANDO, FL. USA 200) having 248 nm central wavelength, 20 ns pulse duration, repetition rate of 20 Hz and with rectangular beam size of 1.1 cm × 0.7 cm was utilized for ablation of Al samples. The 1.2 cm × 1.2 cm square shaped Al samples (commercially available 99.99%) were utilized as target. The Al samples were grinded, polished and cleaned ultrasonically with acetone (Merck Ltd Shuguangxili, Beijing China) for 30 min prior to laser treatment. These polished samples were mounted on target holder placed in vacuum chamber (locally manufactured with good out gassing properties), made of stainless steel, evacuated at a base pressure of about 10^−3^ mbar. Convex lens (Edmund Optics Inc., Singapore) with 50 cm focal length, was used to focus laser beam on the target surface at 90° with respect to target surface. Targets were exposed to 100 laser pulses with repetition rate of 20 Hz, for both ambient environments (non-reactive Ar and reactive O_2_). The targets of Al were exposed for four laser pulsed energies of 95, 110, 125 and 140 mJ with corresponding laser fluence values of 0.86, 1, 1.13 and 1.27 J^·^cm^−2^, in each environment at a filling pressure of 100 torr. The laser fluence was calculated by using the following formula,
(1)$$Fluence=\frac{E}{A}$$

Here, *E* (mJ) is energy of incident laser beam and *A* (cm^2^) is area or the beam spot size for single pulse irradiation and was about 0.11 cm^2^ in present case.

Ablation threshold of Al can be calculated by following equation [29],
(2)$$\;F_{th}=\rho L_{v}a^{1/2}t_{p}{}^{1/2}$$
where ρ  is density of target, specific heat of vaporization/mass is represented by $L_{v}$, the thermal diffusivity of target is a and duration of pulse is represented by *t_p_*.

The value of thermal diffusivity can be calculated by Equation (3),
(3)a = K/ρC

Here, K is the thermal conductivity (79.5 W/mK) and C is the specific heat (0.900 J/gK) and ρ = 2.7 g cm^−3^ of Al target. By using these values in Equation (3), we get value of a = 0.32716 s^−1^ cm^2^. After that, by using ρ = 2.7 g cm^−3^, *L_v_* = 10,874 J g^−1^, a = 0.32716 s^−1^ cm^2^ and *t_p_* = 1.8 × 10^−8^ s in Equation (2), we get the value of ablation threshold fluence for Al that was about 2.25 J^·^cm^2^.

After laser irradiation the ablated targets were characterized by using various characterization tools. SEM (JEOL JSM-6480 LV, JEOL Ltd., Tokyo, Japan) and AFM (Atomic Force Microscopy, Shimadzu AFM-9500J, Shimadzu Corp., Kyoto, Japan) were utilized to investigate the Surface morphology of the laser-irradiated Al targets. X’Pert PRO-MPD X-ray diffractometer (Malvern Panalytical, Malvern, UK) working in θ–θ mode, with 40 mA current and 40 kV voltage was used for crystallographic, dislocation density, residual stress measurements and for study of variation in composition of laser-ablated Al targets. Meanwhile, the EDS (S3700N) analyzer (Hitachi, Tokyo, Japan) was used for chemical compositional analysis. Micro Raman Spectrometer (Dongwoo Optron, MSS-400A, Gwangju, Korea) was used to identify the formation of new bonds in Al after exposure in O_2_ ambient. Nano-indentation Tester (CSM Instruments, NHT + MHT, Needham Heights, MA, USA) was used for the measurement of nanohardness of laser-ablated Al.

## 3. Results and Discussions

Figure 1a–d represent SEM images revealing formation of ripples or Laser Induced Periodic Surface Structures (LIPSS) after irradiation of Al under 100 torr pressure of Ar for various laser fluences. Increase in fluence from 0.86 to 1.27 J^·^cm^−2^ results in decrease in average periodicity of ripples from 22 to 14 µm as shown in the Figure 1a–d. With increasing laser fluence, the ripples merge together and this fusion is significantly enhanced at the edges. With increasing fluence, enhanced absorption of energy is responsible for decreased tendency in the periodicity of these ripples. This enhanced energy deposition is responsible for profound plasma surface interactions, thermal effects, melting, re-solidification, spatial in-homogeneities and surface tension gradients, and consequently causes the merging of individual periodic structures [30].

It is also observed that the periodicity of these laser-generated LIPPS (14 to 22 µm) is significantly greater than the laser wavelength (248 nm). If the damage threshold of the target material is less than the laser fluence, the laser will heat the target further than the vapor phase, which results in a wavy melt pool responsible for the formation of this kind of structures after irradiation [31]. Surface tension increases for molten Al, which causes the convective flow from hotter to colder region at the surface [32].

Bonse et al. [33] discussed some intra (during single laser pulse) and inter (for successive laser pulses) pulse feedback processes for the formation of ripples on laser-irradiated surfaces. The intra-pulse effects can be related to spatial in-homogenous absorption, self-trapped excitation/excitations of transient defect states, transient variation in optical properties of irradiated targets and nonlinear effects. While the inter-pulse effects include ablation and alterations in surface morphologies, hydro-dynamical flow of melted material, variation in material surface structures in between amorphous and crystalline states affecting optical constants. Further, the chemical reactions of ambient environment, like oxidation and permanent defects generation by incubation causing reduction in ablation threshold [34] or self-organized surface erosions and diffusion of atoms, are viable. LIPSS generation on irradiated Al targets can also be discussed on grounds of Kelvin–Helmholtz instabilities [31]. In the present case, the irradiated Al surface is behaving as a fluid, while the expansion of plume with high pressure will act as the source of wind. Ripples with size in micrometer range are developed on the laser-irradiated target due to the re-solidification of splashed laser-induced surface waves [30,31].

Magnified SEM images of irradiated Al at the central ablated area are shown in Figure 2a–d. Fibrous structures like nanoscale ripples appear on the irradiated area of Figure 2a, along with droplets and cavities. Heating induced by laser causes thermal desorption, thermal ablation and explosive boiling after melting of the irradiated area which results in cavity formation [35] and the hydro-dynamical effects are responsible for the formation of droplets. With the increase of fluence, the fibrous structures are transformed to micro-cones exhibited in Figure 2c,d. It shows that the threshold value for the development of conical microstructures is greater than that required for the formation of ripples [36]. The formation of cones originates from the concurrent action of several deformation phenomena, like ablation, melting and evaporation of the material [37]. Properties of material (heat capacity, thermal conductivity, thermal diffusivity), laser parameters (fluence, repetition rate, number of pulses) and the ambient environment (reactive or non-reactive) affects each deformation mechanism [37]. Another reason for cone formation after multiple pulse irradiation of the target will be the erosion resistance generated by the defect generation and surface segregation that results in preferential evaporation of the target surface, which in turn results in the formation of cones [38].

Figure 2c,d also show decrease in density of cones for increase of laser fluence from 1.13 to 1.27 J^·^cm^−2^. It also exhibits that the cones become less distinct. Thermal effects due to enhanced laser fluence destruct the previously formed well-defined structures. Figure 2c also reveals the presence of a large number of cavities at fluence value of 1.13 J^·^cm^−2^. Density of cavities gets reduced for maximum laser fluence (1.27 J^·^cm^−2^) and is revealed in Figure 2d. Melting due to enhanced localized ablation is accountable for reduction in the density of cavities [39].

Figure 3a–d reveal the SEM images of overall ablated area of Al targets exposed to 100 pulses of Excimer laser, under 100 torr pressure of O_2_ for various laser fluences. Small-scale diffused ripples are observed at laser fluence of 0.86 J^·^cm^−2^ (Figure 3a).

Distinguished and well-defined spiral-shaped LIPPS with an average periodicity of 28 µm are observed with increase in laser fluence from 1 to 1.27 J^·^cm^−2^ as shown in Figure 3b–d. Figure 3b,c represents the growth of imprinted name of Allah (ﷲ), the God in Arabic, while it is repeated twice in Figure 3d in the form of spiral-shaped LIPSS. For all fluences periodicity of the spiral shaped LIPPS remains almost constant. Redeposited material and cavities are also observed along the channels. Melting of the material due to laser induced heating in O_2_ environment causes formation of oxides and is also responsible for the redeposition of the ablated material. The convective instabilities, gradient of temperature spatial in-homogeneities, tension of the surface during resolidification, redeposition, oxide formation and enhanced exothermic reactions are accountable for the development of spiral shaped structures [40]. Wave-like ridges are also observed at the boundary of the ablated area. The piston mechanism can be related to the ablation mechanism and formation of ridges, in which laser produced plasma and the recoil pressure of the ejected material moves a molten pool layer to boundaries of spot of laser beam which coagulate in the shape of ridges.

Enlarged views (at higher magnification) of central ablated region after irradiation in ambient environment of O_2_ for 100 laser pulses and for different laser fluences at pressure of 100 torr are shown in Figure 4a–d. Wave-like structures or small-scale ripples along with wrinkles and canals are observed in Figure 4a. Fibrous structures or small scale ripples are also presented in Figure 4b–d.

Comparison of both non-reactive (Ar) perceived in Figure 1 and Figure 2, and reactive (O_2_) environment shown in Figure 3 and Figure 4, presents considerable differences for the ablated areas. In the case of non-reactive environment (Ar), the ablated area reveals prominent ripple formation (Figure 1), droplets and fibrous structures. On the other hand, for reactive environment (O_2_), less distinct spiral-shaped ripples along with redeposited material are observed (Figure 3).

The formation of distinct and well-defined ripples on Al targets after irradiation in non-reactive (Ar) ambient as compared to the reactive (O_2_) environment, is basically due to enhanced electronic cascade growth in Ar compared to the O_2_ environment [41]. During cascade growth of electrons, the energy loss can be calculated by measuring the *E*/M ratio [41], where *E* is the first ionization energy of the gas and M is mass of the background gas. *E*/M value for O_2_ is 0.43 and *E*/M value for Ar is 0.39, which shows that cascade growth for Ar will be higher than in O_2_ [41]. Enhanced electronic cascade growth escorts to plasma are much higher in temperature and density in the Ar environment as compared to the ambient environment of O_2_, which is accountable for efficient energy transfer to the sample surface. This is the major reason for the appearance of more distinct and prominent ripples in Ar than in O_2_.

Figure 5 shows the AFM surface topographical image of untreated Al surface for scan area 5 µm × 5 µm with surface roughness value of about 10 nm.

Figure 6a,b depicts the AFM surface topographies of Al targets of 5 µm × 5 µm scan area, exposed to 100 laser pulses at a pressure of 100 torr for 0.86 J^·^cm^−2^ fluence in Ar and O_2_ environments respectively. The surface roughness value of irradiated Al in Ar environment is about 87 nm, while in O_2_ environment it is about 44.88 nm. These results show that surface roughness value in both ambient environments is higher than the untreated Al surface. KrF Excimer laser system generate the pulses in UV spectral range, causing abrupt material removal from target surface with less penetration of light into sublayers of material [42]. So, this laser system is most suitable for etching and growth of surface structures on irradiated targets and generation of permanent plastically deformed materials [43]. These deformations cause the enhancement in surface roughness and generating the anti-reflective properties on the surface and enhance the optical absorption [44]. This enhancement in the optical absorption develops black metal effects on ablated targets and makes the material useful in the underwater marine applications.

Figure 6a shows the presence of protrusions, bumps and cones of height of 250 nm for Ar environment, whereas an average height of 150 nm is revealed for the bumps formed after irradiation in O_2_ environment, as shown in Figure 6b. Non-uniform and protruded surfaces are generally termed as bumps, having high electric fields and defect densities in the regimes near to surface. Many researchers have found the presence of micro/nano bumps on metals and nonmetals [45,46]. Surface aggregation of defects after diffusion of gasses can generate the bumps on irradiated surface. Relaxation of compressive residual stresses and enhancement in local volume is also one of the basic reasons for protruded surfaces, cones and bumps formation [47]. Formation of bumps can also be attributable to micro-explosions owing to the pressure waves generated by melting of the surface [47].

Table 1 shows the variation in elemental composition (EDS data) of unexposed and exposed Al targets, exposed to 100 laser pulses, at laser fluence of 0.86 J^·^cm^−2^ and under pressure of 100 torr in Ar and O_2_ environments. EDS point analysis data was collected from the central ablated areas of the samples and the data given in Table 1 is with percentage error of ±5%. Variation in Al content owing to inclusion of Ar and Atomic Oxygen (O) into the Al surface after irradiation in both ambient environments is shown in the table. Small amount of O observed in the untreated Al is attributable to the formation of oxide due to chemical reaction of Al with the ambient environment when placed in air before experiment. During ablation in ambient environment of Ar, complete removal of oxides is observed. A small amount of Ar (1 atomic%) is seen, which is attributable to the diffusion of Ar atoms interstitially into the Al target. Being inert gas, only highly energetic Ar ions (small in number) can diffuse into the Al surface. Meanwhile, during irradiation in ambient environment of O_2_, Al concentration is decreased from 99 (atomic %) to 65 (atomic %) owing to penetration of oxygen atoms into the surface (varies from 1 to 35 atomic %). It has been suggested that the recoil pressure of plasma plume species, may also promote the diffusion and incorporation of O_2_ gas atoms into the molten Al surface [48]. Dissociation of molecular oxygen (O_2_) to atomic oxygen (O) takes place more efficiently during laser irradiation. As the adsorption capacity of the target is higher for O than molecular oxygen (O_2_), this therefore results in significantly enhanced diffusion of O. XRD results supports the EDS results, where enhanced diffusion of oxygen results in peak broadening.

During EDS analysis, the electron beam interacts with the sample (Al in present case) and the characteristic X-rays are emitted, analyzed and give results for EDS analysis. Here we can calculate the penetration depth of electrons in the Al surface by using the Kanaya–Okayama model [49] (https://www.globalsino.com/EM/page4795.html accessed on 6 June 2021)
(4)$$R=\frac{0.0276\;A\;E_{0}^{n}}{\rho Z^{0.89\;}}$$

where, for Al targets,

*R*—Penetration Depth*A*—Atomic Weight (g/mol) = 26.98153*n*—A constant an E_0_ > 5 keV so its value is 1.67*E*_0_—Beam Energy (keV) = 10 keV*Z*—Atomic number = 13ρ—Density = 2.7 g/cm^3^.

Now, replacing all these values in Equation (4), we get the value of penetration depth, which comes out to be 1.315 µm for an electron beam in to the Al target that is surface-to-bulk penetration depth.

Laser beam penetration depth can be calculated from the following formula [50],
(5)$$\delta_{p}=\frac{1}{\propto_{0}}$$

Here, penetration depth is represented by $\delta_{p}\;$, and $\propto_{0}$ is the absorption co-efficient of 0.248 µm (248 nm) laser beam in Al, which is about 1.4711 × 10^6^/cm [51].

By using this value in Equation (5) we can find the penetration depth of KrF laser beam into the Al target, which is about
0.6797 µm.

Here, we can see that the penetration depth of the electron beam during EDS analysis is higher in Al as compared to the laser beam during the laser ablation experiment. The EDS analysis results are given in Table 1. We have seen that changes in chemical composition (comparison of three samples) are still observed.

After laser irradiation, the vaporization of target takes place, which causes the generation of high temperature and high-pressure plasma [52]. This plasma during expansion introduces pressure and compressive waves in giga pascal (GPa) range, which causes propagation of shock waves on and into (bulk) the irradiated target surface, due to which the material gets plastically deformed due to enhancement in peak pressure. When it becomes higher than dynamic yield stress/strain, the generation of compressive stresses and material’s resistance to fatigue corrosion and cracking get enhanced (confirmed from the EDS and XRD analyses, too) [21]. This causes the generation of surface structures on the surface and also variation in the chemical composition on the surface as well as into the bulk material. That is why we obtained variation in chemical composition in EDS results, due to presence of shock effected zones into the bulk material after laser irradiation.

Figure 7 shows XRD patterns of untreated and irradiated Al targets in Ar ambient, exposed to 100 pulses for various laser fluences at pressure of 100 torr. The inset of Figure 7 presents the magnified view of Al (200) peak, which clearly reveals the variation in the intensity of peak and peak position for both untreated and irradiated targets. During irradiation of Al with laser, the target receives a large amount of energy, which increases the surface temperature. This rise in temperature results in permanent distortion of lattice and d-spacing variations due to the difference in thermal expansion coefficients and differences in inter atomic distances [53].

The surface temperature after KrF Excimer irradiation can be calculated by using the following relation [54],
(6)T=T0+2I01−R0K α0tpπ^1/2

Here, room temperature is denoted by *T*_0_, *I*_0_ is the laser beam intensity, surface reflectivity is represented by *R*_0_ (0.926 for 248 nm laser beam in Al), *K* is the thermal conductivity (79.5 W/mK), the absorption co-efficient is represented by $\alpha_{0}$ (1.4711 × 10^6^/cm) and *t_p_* is pulse duration of laser beam (1.8 × 10^−8^ s). By using these values in the above equation, we can find the value of surface temperature that comes out to be 1.549 × 10^3^ K for minimum fluence value (0.86 J^.^.cm^−2^) and 2.26 × 10^3^ K for maximum fluence value of 1.27 J^·^cm^−2^.

The XRD spectrum of untreated Al shows the existence of (200), (111), (220) and (311) plane orientations (Pattern No. 01-089-4037). During treatment in the Ar environment, new phases are not observed. However, variation in the peak intensity, crystallinity, residual strains and residual stresses are observed for Al (200). The crystallinity was calculated from the peak broadening of diffraction plane Al (200) using Scherrer’s formula [55].


Crystallite size (D) = 0.9λ⁄(β cosθ)
(7)

where D is crystallite size, λ is X-ray’s wavelength (1.5406 Å), β is full width at half maximum and θ is Bragg’s angle of diffraction.

Residual strain values are evaluated by using relation given below [56].


Strain(ɛ) = (d − d_o_)/d_o_(8)

where d is the observed and d_o_ is the standard spacing between the crystalline planes. The induced stresses σ are defined as


σ = ɛE
(9)

where E is the Young modulus, equal to 69 GPa for Al [57].

Figure 8a–c depicts the variation in crystallite size, peak intensity, Full Width at Half Maximum FWHM, residual strain as well as residual stress for Al (200) after ablation in Ar ambient, for 100 laser pulses and various laser fluences at pressure of 100 torr. Untreated Al shows crystallite size of 209 nm for Al (200) reflection plane. Figure 8a represents the variation in crystallite sizewith increase in laser fluence, whereas the variation in crystallite size and FWHM with the enhancement in laser fluence is shown in Figure 8b. Both peak intensity and crystallite size increase with the increase in laser fluence. On the other hand, reduction in FWHM of Al (200) diffraction plane is perceived with an increase in laser fluence up to 1.27 J^·^cm^−2^.

After irradiation with laser variation in intensity of peak is correlated to the variations in size of crystallites, residual strain and residual stress on target surface. Variation in strain, stress and lattice distortions are attributable to changes in d-spacing, which are attributable to the temperature and cooling conditions, thermal expansion coefficient, variations in inter atomic distances, boundary and interstitial diffusion among the surface layers [58]. Observed enhancement in intensity of peak is attributable to reduction in peak broadening and increase in crystallite size of Al (200). After irradiation with laser, shift of diffraction planes to the unstrained positions [59], lattice imperfections and crystal defects causes the increase in intensity of Al (200) peak [60,61]. Development of thermal shocks on the surface of target during irradiation also plays a momentous role in the growth of crystals. Lattice defects created by the implanted ions into the lattice and laser-induced thermal shocks can cause residual stress variation. Thermal shocks induced by end of laser pulses generally cause tensile residual stresses, whereas ion implantation results in compressive residual stresses [62]. Compressive residual stress observed for laser fluence of 0.86 J^·^cm^−2^ is due to small crystallite size, which offers more boundary area and slows down the slip motion and results in enhanced strength of material by offering the compressive residual stress (enhanced strength can be verified from hardness results). During the irradiation process, sufficient energy is brought to target surface. This increases surface temperature of the target, which causes lattice distortion and reduced crystallite size in return causes the generation of compressive residual stresses on the irradiated target [63]. When the target material is exposed to the laser beam, a thin surface layer of the target material vaporizes and expands perpendicular to the target surface and generates a huge pressure in range of GPa over the surface [64,65]. Jelani et al. [66] in his research reported generation of pressure in GPa range for zirconium irradiated with KrF Excimer laser, which causes the generation of compressive residual stresses.

The relation between residual strain and residual stress with variation of laser fluence is shown in Figure 8c. Increase in the fluence value from 0.86 to 1 J^·^cm^−2^ causes relaxation in the compressive residual stress. Further increase in fluence up to 1.27 J^·^cm^−2^ transforms this compressive residual stress to a tensile one. Increase in crystallite size due to laser induced thermal shock with increase in laser fluence is responsible for the relaxation in compressive residual stresses and their transformation to the tensile residual stresses [45]. Relaxation of compressive stresses and their transformation to tensile stresses with increase in laser fluence can also be confirmed from the peak shift towards lower angular position (inset of Figure 8). These results are well correlated with surface features shown in Figure 2a–d. The formation of cones and enhancement in size and density of cavities with the increase in laser fluence is evident for the relaxation and transformation of compressive stresses to tensile ones.

Figure 9 shows XRD patterns of untreated and laser-irradiated Al targets in O_2_ environment, exposed to 100 laser pulses, at pressure of 100 torr, at various laser fluences.

The XRD spectrum demonstrates the conversion of Al (200) phase to Al_2_O_3_ (202) (Pattern No. 00-001-1296). Newly emerging peaks of Al_2_O_3_ (222), Al_2_O_3_ (533), AlO (200) and AlO (331) [67] are also revealed. The inset of Figure 9 presents the magnified view of Al_2_O_3_ (202) peak, which depicts the variation in the intensity of peak and peak position for both untreated and irradiated targets. The crystallite size of irradiated targets is also calculated for Al_2_O_3_ (202)_._ Young’s modulus for Al_2_O_3_ (at room temperature) is 408.99 GPa [68].

Figure 10a–c depicts the variation in intensity of peak, FWHM, crystallite size, residual strain and residual stress for Al_2_O_3_ (202) after ablation in O_2_ ambient, for various laser fluences and 100 pulses of KrF Excimer laser at pressure of 100 torr. Figure 10a represents, with the increase in laser fluence, the variation in peak intensity, while the variation in crystallite size and FWHM with the variation in laser fluence is shown in Figure 10b. Both peak intensity and crystallite size increase, whereas a decrease in the value of FWHM is observed for increase in fluence values of 0.86 to 1.27 J^·^cm^−2^.

Increase in the intensity of peak with the increase in fluence is attributable to reduction in broadening of the diffraction peak and increase in crystallite size of Al_2_O_3_ (202). During laser irradiation at elevated temperature, crystals grow due to defects and lattice imperfections, which causes the increase in intensity of Al_2_O_3_ (202) peak [60,69]. Due to enhanced heat generation, with increase in laser fluence, solute oxygen atoms segregate on the surface of target and diffuses across the boundary of grains [70] (oxygen diffusion is confirmed by EDS analysis). Motion of diffraction planes to unstrained position may also cause crystal growth [60,69].

Figure 10c shows the relation between residual strain and stress with increasing laser fluence. For minimum laser fluence, compressive residual stress is obtained which relax and alter to tensile residual stress for fluence value of 1.0 J^·^cm^−2^. Increase in fluence value from 1.0 to 1.27 J^·^cm^−2^ results in an increase in tensile residual stress. Increase in fluence causes the peaks to move to lower angular positions, which confirm the relaxation and transformation of compressive residual stress to tensile residual stress. These results are well correlated with surface features shown in Figure 4a–d, where, in Figure 4a, the compact surface shows the presence of compressive stresses and presence of various cavities in Figure 4d, representing the presence of maximum tensile stresses.

The comparison of our present work with our previously reported work related to femtosecond laser ablation of Al in vacuum and O_2_ [34], reveals that more new phases of Al_2_O_3_ and AlO are formed with nanosecond ablation as compared to femtosecond laser ablation in O_2_ ambient. It might be related with the factor that the thermal processes become dominant during nanosecond laser irradiation, causing enhanced oxygen diffusion in the Al matrix that is also confirmed from EDS analysis (35% atomic oxygen diffusion). From the literature it is confirmed that Al_2_O_3_ is most significant phase [25].

Figure 11 shows Raman spectrographs of Excimer Laser-irradiated Al targets under O_2_ ambient for minimum value of fluence (0.86 J^·^cm^−2^).

Formation of oxides on the metallic surface after laser irradiation gives rise to Raman modes. The Raman peaks observed at wave numbers 433, 635 and 742 cm ^−1^ represent polycrystalline γ-A1_2_0_3_ (confirmed from XRD too). The observed peak at 335 cm ^−1^ may show the bending vibrations of AlO_4_ and AlO_6_ groups. The band’s multiplicity shows the deformation splitting and vibrational coupling. Peaks identified at 445 and 487 cm^−1^ correspond to isolated octahedral AlO_6_ groups, whereas the observed peak at 567 cm^−1^ present the condensed AlO_6_ group [71,72].

Nano-indentation analysis was used to analyze the hardness of surface after irradiation with KrF Excimer lasers in Ar ambient for different laser fluences (0.86, 1, 1.13, 1.27 J^·^cm^−2^), as shown in Figure 12a. The nanohardness was measured from five different points on the ablated spot and their average values are presented. The measurement error (standard deviation) was about ±0.03 for each measurement. Elementary principle of hardness investigation is basically to measure the resistance of a material to plastic deformations. Untreated Al target shows a hardness value of 0.19 GPA. After irradiation at 0.86 J^·^cm^−2^ in Ar ambient, it shows hardness of 0.81 GPa, which is 4.26 times more than the hardness of the un-irradiated target. Continuous decrease in hardness up to 0.32 GPa is observed for increase in fluence. This is still 1.68 times higher than the hardness of untreated substrate.

An enhancement in the hardness of Al surface for fluence value of 0.86 J^·^cm^−2^ can be related to the variation in dislocation density and change in crystallite size. Crystallite size of 158 nm is observed for fluence value of 0.86 J^·^cm^−2^, which is less than the crystallite size measured for untreated Al (2 0 0). Grain refinement causes the enhancement in dislocation density and in return hinders the further dislocation motions and causes the enhancement in the surface hardness [73]. The reduction in the grain size causes the enhancement in the grain boundary density, which hinders the dislocation motions and causes the enhanced surface hardness. Enhancement in surface hardness of irradiated Al target can also be related to the generation of shock waves produced due to expansion of high-pressure plasma by end of laser pulse. Interaction of laser beam with the target surface causes to vaporize it to high pressure and high temperature plasma, instantaneously [21]. This laser-ablated plasma from the target surface expands and exerts pressure on the target surface, and inducing compressive waves in Al target in turn causes propagation of shock waves on the sample surface. When shock wave peak pressure becomes greater than dynamic yield stress, then the material surface becomes plastically deformed.

This phenomenon can induce compressive stresses in the surface part, and hence, enhances the material’s resistance to surface failures like its fretting fatigue, fatigue and corrosion stress cracking [21]. Decrease in hardness with further enhancement in laser fluence from 0.86 to 1.27 J^·^cm^−2^ can be attributable to the tensile residual stresses (related to the peak shifting to lower angular positions, Figure 7), causing the enhancement in cavity size (Figure 2) and grain growth (Figure 8b), which in turn causes reduces hardness [73].

After irradiation in O_2_ ambient, variation in hardness for different laser fluences (0.86, 1, 1.13, 1.27 J^·^cm^−2^) is revealed in Figure 12b. For minimum value of fluence, maximum hardness value is achieved, i.e., about 1.83 GPa. That is 9.63 times more than the hardness of untreated substrate and 2.25 times more than the hardness of substrate treated in Ar environment for same laser fluence of 0.86 J^·^cm^−2^. With an increase in the value of fluence up to 1.27 J^·^cm^−2^, hardness decreases up to 0.37 GPa. That is 1.94 times more than the hardness of untreated target and 1.15 times the hardness of targets treated in Ar environment for same fluence of 1.27 J^·^cm^−2^.

Here, we have observed that the hardness value of irradiated targets in O_2_ ambient is higher than the untreated one for all fluence values. This can be attributable to the formation of various phases of AlO, among which the hardest is Al_2_O_3_ phase (confirmed from XRD and Raman analysis) on the irradiated targets. The enhancement in the hardness of Al surface is related to O diffusion to the interstitial sites and lattices, causing enhanced lattice distortions and decrease in size of crystallites and causing enhanced surface hardness [73]. On the other hand, the reduced hardness can be related to crystal growth due to diffusion of O across the grain boundaries and tensile residual stresses causing enhancement in the cavity density and cavity size (Figure 4). From Figure 10, it is observed that the hardness variation directly depends on the crystallite size, O diffusion and presence of residual stresses [35].

## 4. Conclusions

The effect of non-reactive (Ar) and reactive (O_2_) environments on the pulsed laser ablation of Al has been investigated. Micro/nano structuring of KrF Excimer laser-irradiated Al was correlated with laser-produced structural and mechanical changes. In the case of ablation in Ar ambient, the ablated area shows the formation of distinct ripples, small-sized droplets and fibrous structures, whereas after ablation in O_2_ environment diffused ripples along with re-deposited spiral structures and surface wrinkles are exhibited. AFM analysis shows that the surface roughness value to irradiated targets is higher than the untreated samples. AFM analysis also shows the development of protrusions and bumps for both non-reactive and reactive environments caused by segregation of surface defects, micro-explosions caused by pressure waves or due to relation of compressive residual stresses after diffusion of gasses. From EDS analysis, a small amount of Ar (~1 atomic %) in the case of Ar (non-reactive) environment and a comparatively large amount of atomic oxygen (~35 atomic %) diffusion in the case of O_2_ (reactive) environments is observed. This O diffusion is also confirmed from the XRD results, which show the broadening of peaks due to O ions diffusion and formation of new peaks. X-ray diffraction analysis of the samples treated in Ar environment exhibits that no new phases are formed, while new phases of oxides (Al_2_O_3_, AlO) are produced in the case of O_2_ treatment. Raman analysis shows the presence of γ-A1_2_0_3_ in AlO_4_ and AlO_6_ vibration groups in O_2_ ambient, which supports the XRD and EDS results.

In this study, a strong correlation is observed between the surface structuring, chemical composition, residual stress variation and variation in surface hardness of Al after ablation in both ambient (Ar, O_2_). During laser–material interaction, in presence of background gas, special confinement and shielding effect slow down the expansion of the plasma plume, in which several thermal phenomenon generated by laser heating through the gas–solid interface, in the case of non-reactive environment (Ar), results in variation in micro-strains and stresses, which causes the development of a variety of surface features and variation in surface hardness. In the case of reactive environment (O_2_), improved reactivity and interplay among the plasma species of ambient environment and laser energy deposition causes the formation of Al_2_O_3_ (alumina) with high mechanical strength, which makes it useful in the field of process and aerospace industry as well as in engineering. Nanohardness measurement shows that hardness variation depends upon the variation in crystallite size. It is also observed from this analysis that enhancement in hardness in O_2_ (reactive) ambient is higher as compared to Ar (non-reactive) ambient.

## Figures and Tables

**Figure 1 materials-14-03671-f001:**
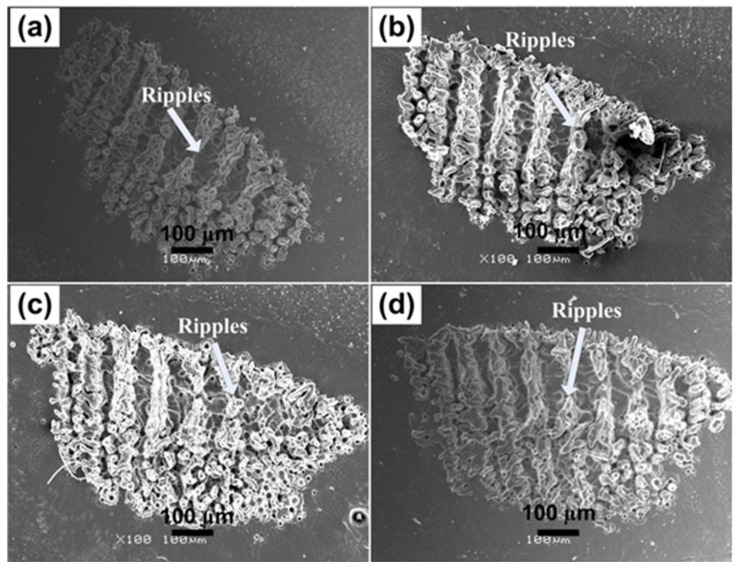
SEM images revealing the surface morphology of Al exposed to 100 pulses of ns laser at wavelength of 248 nm, pulse duration of 18 nm and repetition rate of 20 Hz at various fluences of (**a**) 0.86 J^·^cm^−2^, (**b**) 1 J^·^cm^−2^, (**c**) 1.13 J^·^cm^−2^ and (**d**) 1.27 J^·^cm^−2^ under an ambient environment of Ar at a pressure of 100 torr.

**Figure 2 materials-14-03671-f002:**
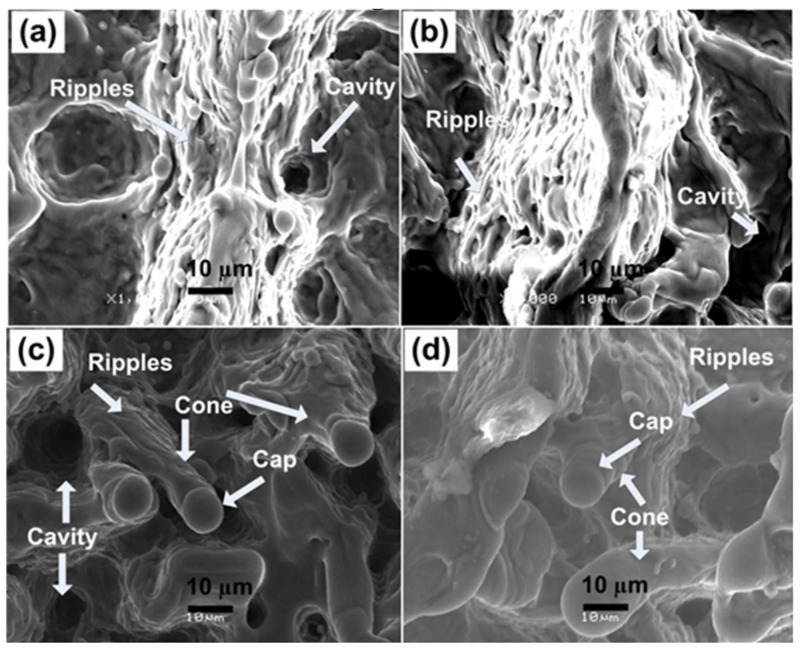
Magnified SEM images of Figure 1a–d at the central ablated area showing the variation in the surface morphology of Al exposed to 100 pulses of ns laser at wavelength of 248 nm, pulse duration of 18 ns and repetition rate of 20 Hz at various fluences of (**a**) 0.86 J^·^cm^−2^, (**b**) 1 J^·^cm^−2^, (**c**) 1.13 J^·^cm^−2^ and (**d**) 1.27 J^·^cm^−2^ in an ambient environment of Ar at a pressure of 100 torr.

**Figure 3 materials-14-03671-f003:**
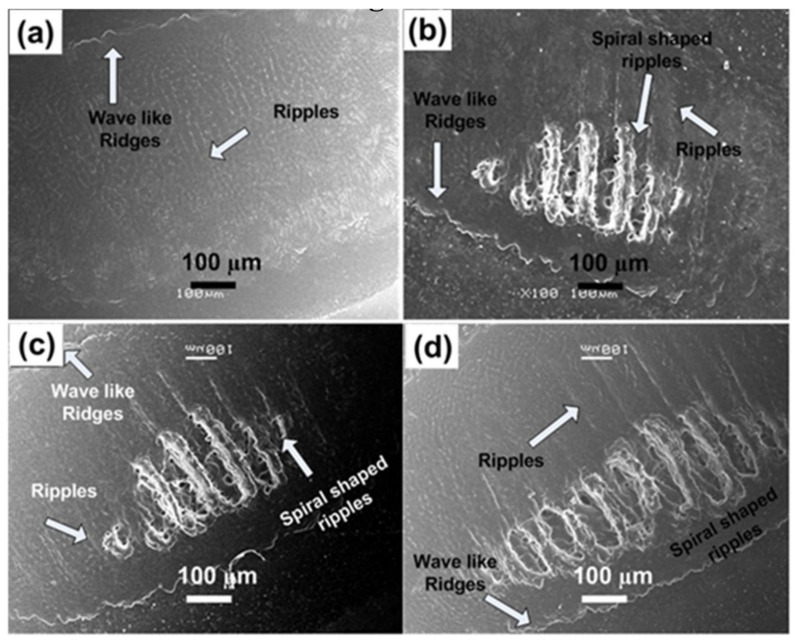
SEM images revealing the surface morphology of Al exposed to 100 pulses of ns laser at wavelength of 248 nm, pulse duration of 18 ns and repetition rate of 20 Hz at various fluences of (**a**) 0.86 J^·^cm^−2^, (**b**) 1 J^·^cm^−2^, (**c**) 1.13 J^·^cm^−2^ and (**d**) 1.27 J^·^cm^−2^, under an ambient environment of O_2_ at a pressure of 100 torr.

**Figure 4 materials-14-03671-f004:**
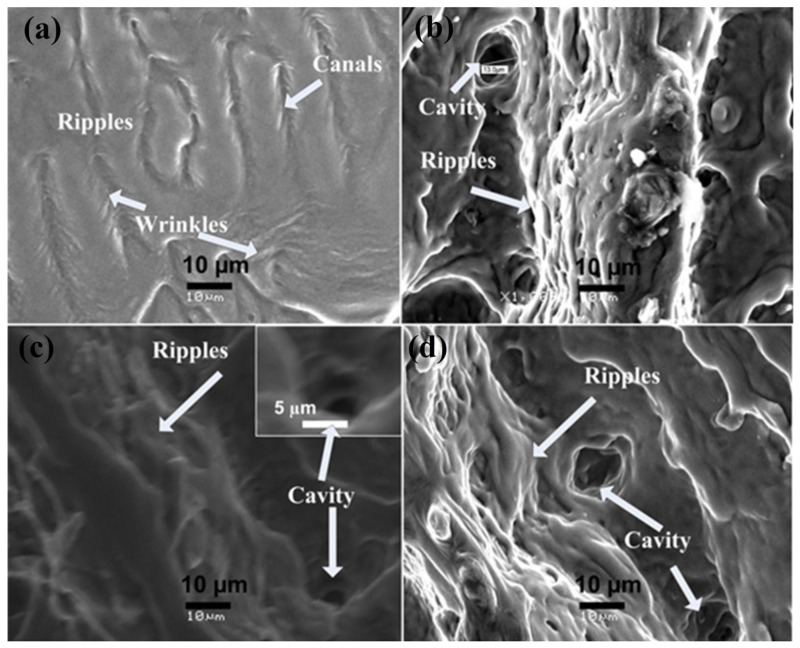
Magnified SEM images of Figure 3a–d, representing the surface morphology of the central ablated area of Al exposed to 100 pulses of ns laser at wavelength of 248 nm, pulse duration of 18 ns and repetition rate of 20 Hz at various fluences of (**a**) 0.86 J^·^cm^−2^, (**b**) 1 J^·^cm^−2^, (**c**) 1.13 J^·^cm^−2^ and (**d**) 1.27 J^·^cm^−2^, under an ambient environment of O_2_ at a pressure of 100 torr.

**Figure 5 materials-14-03671-f005:**
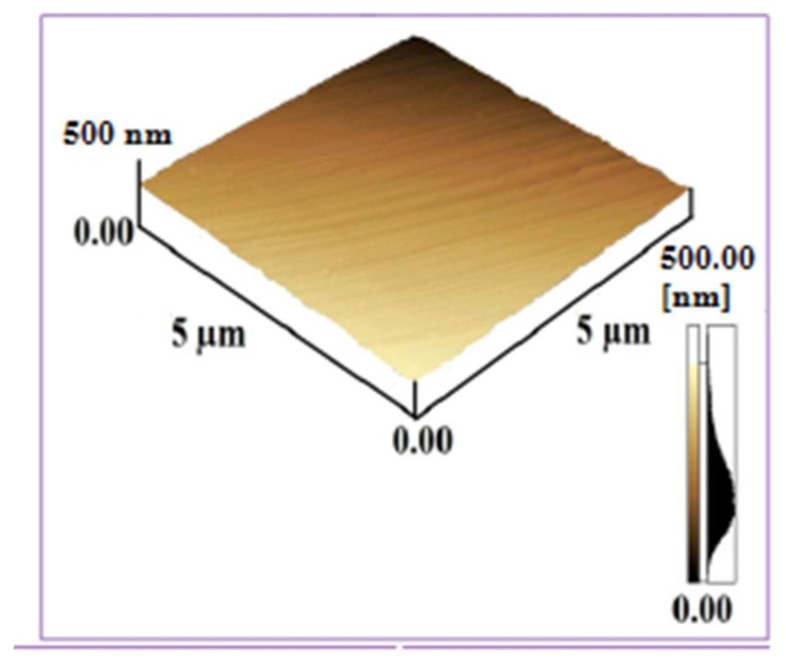
AFM surface topographical image of untreated Al surface for scan area 5 µm × 5 µm.

**Figure 6 materials-14-03671-f006:**
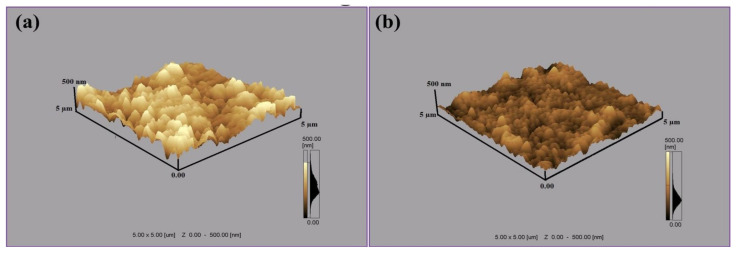
AFM surface topographical images of Al targets exposed to 100 pulses of Excimer laser at a wavelength of 248 nm, pulse duration of 18 ns and for a laser fluence of 0.86 J^·^cm^−2^, in ambient environments of (**a**) Ar and (**b**) O_2_ at a pressure of 100 torr.

**Figure 7 materials-14-03671-f007:**
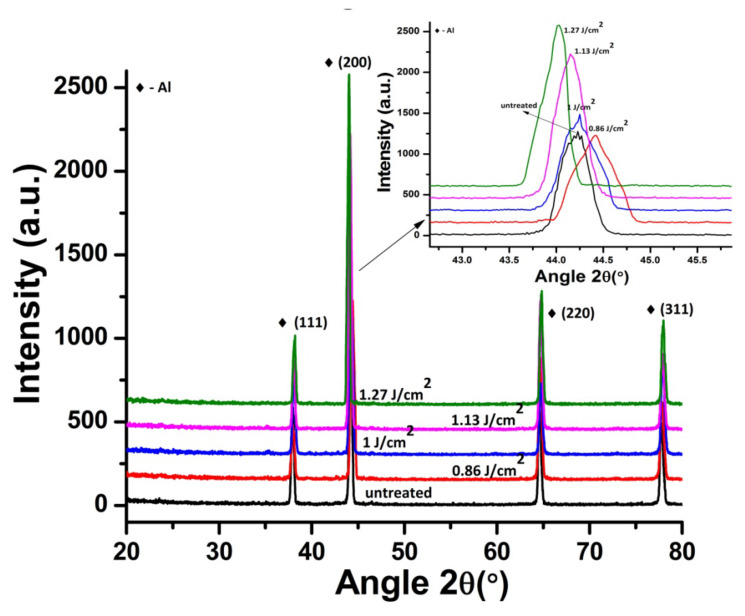
XRD data of un-irradiated and irradiated Al exposed to 100 pulses of ns laser in Ar environment at pressure of 100 torr, at various fluences of 0.86, 1, 1.13 and 1.27 J^·^cm^−2^. Inset of Figure 7 reveals the enlarged view of Al (200) peak, showing variation in peak position and peak intensity.

**Figure 8 materials-14-03671-f008:**
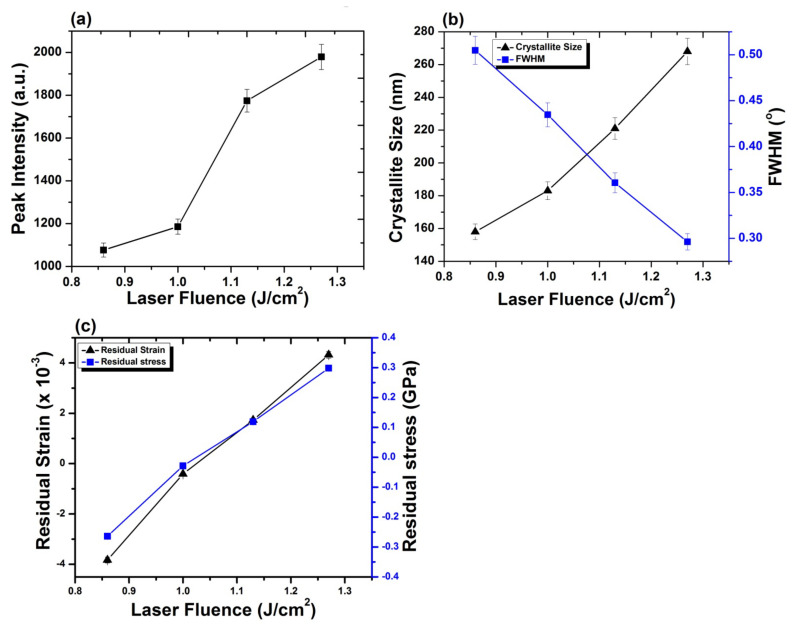
(**a**) The variation in peak intensity, for Al (200) orientation plane, with variation of laser fluence from 0.86 to 1.27 J^·^cm^−2^, under Ar ambient for 100 laser pulses at pressure of 100 torr; (**b**) the variation in crystallite size, FWHM; and (**c**) relation between residual strain and residual stress with variation of laser fluence.

**Figure 9 materials-14-03671-f009:**
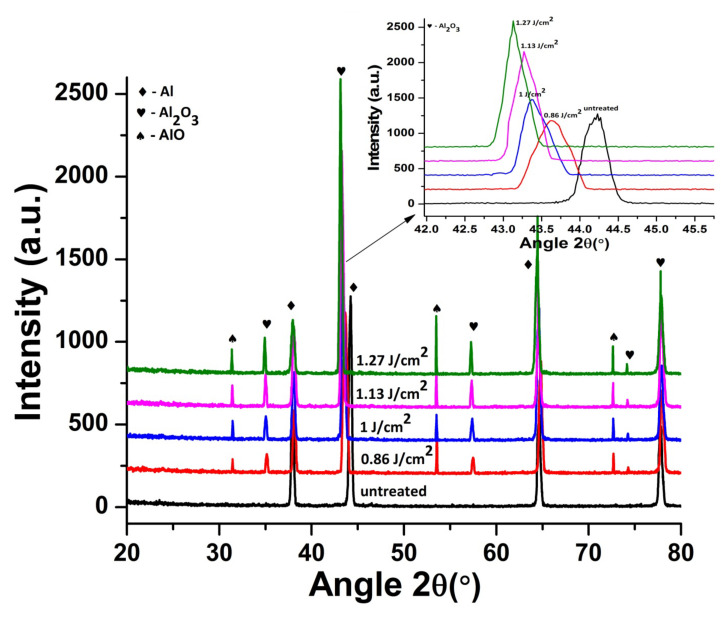
XRD data of un-irradiated and irradiated Al exposed to 100 pulses of ns laser under O_2_ environment at pressure of 100 torr, at various fluences of 0.86, 1, 1.13 and 1.27 J^·^cm^−2^. Inset of Figure 9 shows the enlarged view of Al (200) peak showing variation in peak position and peak intensity.

**Figure 10 materials-14-03671-f010:**
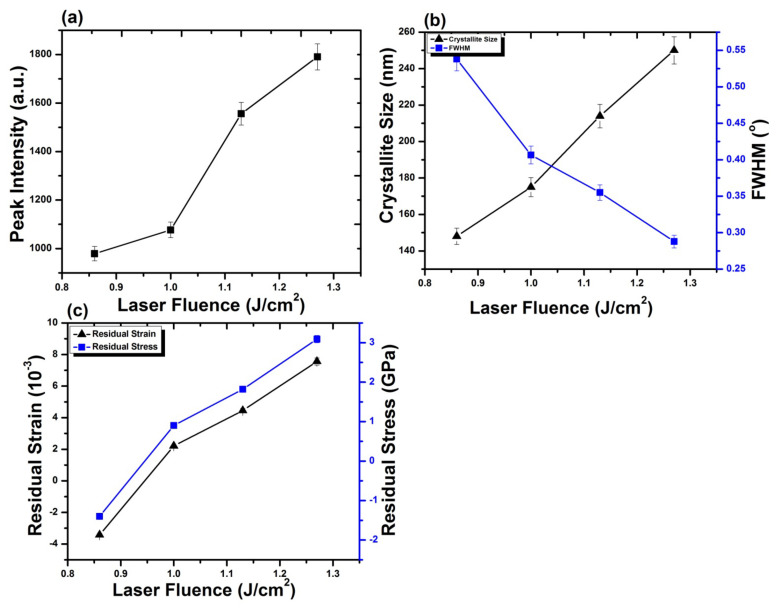
(**a**) The variation in peak intensity for Al_2_O_3_ (202) orientation plane with variation of laser fluence from 0.86 to 1.27 J^·^cm^−2^, under O_2_ ambient for 100 laser pulses at pressure of 100 torr. (**b**) The variation in crystallite size, FWHM and (**c**) relation between residual strain and residual stress with variation of laser fluence.

**Figure 11 materials-14-03671-f011:**
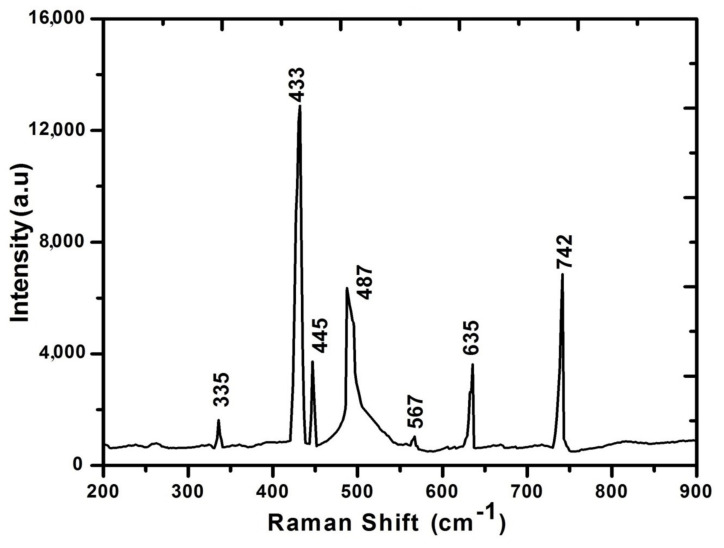
Raman spectrograph of laser-irradiated Al exposed to 100 pulses of ns laser at fluence value of 0.86 J^·^cm^−2^, under O_2_ environment at 100 torr pressure.

**Figure 12 materials-14-03671-f012:**
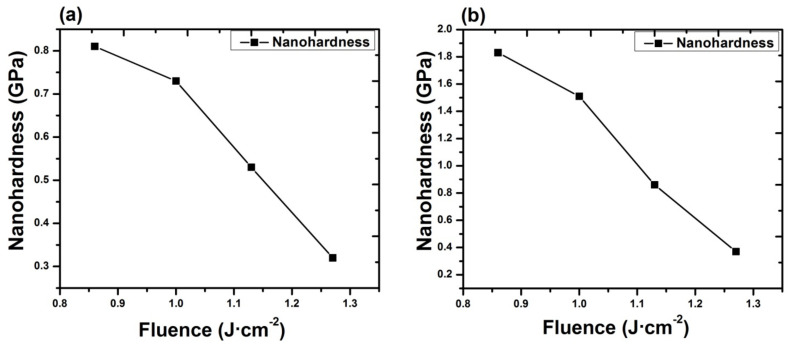
The variation in nanohardness of laser-irradiated Al exposed to 100 pulses of ns laser, at different fluences ranging from 0.86 to 1.27 J^·^cm^−2^, (**a**) under Ar, (**b**) under O_2_ environment at 100 torr pressure. The measurement error (standard deviation) was about ±0.03 for each measurement.

**Table 1 materials-14-03671-t001:** An EDS analysis of the un-irradiated and nanosecond laser-irradiated Al targets, at a laser fluence of 0.86 J^·^cm^−2^, under 100 torr pressure of Ar and O_2_.

Elements	Untreated (atomic %)	In Argon (atomic %)	In Oxygen (atomic %)
Al	97	99.0	65
Ar	-	1	-
O	3	-	35

## Data Availability

All the data is available within the manuscript.

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
