# Peer review of "Study of Micro/Nano Structuring and Mechanical Properties of KrF Excimer Laser Irradiated Al for Aerospace Industry and Surface Engineering Applications"

_materials, 2021, doi:10.3390/ma14133671_

Round 1

Reviewer 1 Report

The manuscript by Kalsoom et al. reported the microstructuring of Al by KrF excimer laser and its influence on the mechanical properties of the material. These results may be of interest to readers. However, several issues need to be clarified before manuscript acceptance.

1) The values of surface roughness before and after laser treatment have to be presented.

2) What is the beam size at the target surface, and how was fluence calculated? What is the ablation threshold?

3) Why this value of 100 pulses was chosen?

4) It is necessary to provide electron energies K at EDS measurements to understand the penetration depth. Moreover, it is strongly recommended to study surface-to-bulk elemental composition by measurements with various K to understand the size of the modified region.

5) Some estimations of surface temperature during irradiation need to be provided.

6) Figure 7 caption needs to be improved.

7) For nanohardness analysis, the measurement error has to be introduced and images of typical imprints of the indenter.

Author Response

Dear Editor,

We have revised the manuscript according to the reviewer’s comments.

All issues raised are addressed point by point and a file dealing with answers to reviewer’s comments is attached separately.

All changes are incorporated in the manuscript also.

 All the answers to reviewer’s comments are given in RED and all changes made in the manuscript according to reviewer’s comments are also highlighted with RED color.

Regards

Nisar Ali

Comments and Suggestions for Authors

The manuscript by Kalsoom et al. reported the microstructuring of Al by KrF excimer laser and its influence on the mechanical properties of the material. These results may be of interest to readers. However, several issues need to be clarified before manuscript acceptance.

  • The values of surface roughness before and after laser treatment have to be presented.

Ans: The surface roughness values and discussion related to it is inserted in the manuscript.

  • What is the beam size at the target surface, and how was fluence calculated? What is the ablation threshold

Ans: The targets of Al were exposed for four laser pulsed energies of  95, 110, 125 and 140 mJ with corresponding fluence values of  0.86, 1, 1.13, and 1.27 Jcm−2, in each environment at a filling pressure of 100 torr. The laser fluence was calculated by using the following formulae,

                                        (1)

Here, E (mJ) is energy of incident laser beam and A (cm2) is area or the beam spot size for single pulse irradiation and was about 0.11 cm2 in present case.

            Ablation threshold of Al can be calculated by following equation (M. H. Iqbal 2015).

                                                   (2)

Where is density of target, specific heat of vaporization / mass is represented by , the thermal diffusivity of target is  and duration of pulse is represented by tp.

The value of thermal diffusivity can be calculated by equation (3)

 = K/ C                                            (3)

Here K is the thermal conductivity (79.5 W/mK) and C is the specific heat (0.900 J/gK) Here  = 2.7 g cm−3 of Al targe. By using these values in equation (3) we get value of  = 0.32716 s−1 cm2. After that by using  = 2.7 g cm−3, Lv = 10874 J g−1,  = 0.32716 s−1 cm2 and tp = 1.8 × 10−8 s in equation (2) we get the value of ablation threshold fluence for Al that was about 2.25 J/cm2.

  • Why this value of 100 pulses was chosen?

Ans: Initially the test experiment was performed for various numbers of pulses for maximum fluence value and observed that for less number of pulses the heat accumulation effects were minimal. While for higher number of pulses the heat accumulation effects were quite dominant and were not good for surface structuring and we get best results for 100 number of pulses. That’s why we have chosen 100 pulses for varying fluence value in different ambient. Basically the aim behind present work is to observe the modification of surface and structural properties of Al after pulsed laser ablation in Ar and O2 for various fluence values (for fixed number of laser pulses) and is to modify broad range application based functional as well as physical and chemical modifications of laser irradiated Al targets.

4) It is necessary to provide electron energies K at EDS measurements to understand the penetration depth. Moreover, it is strongly recommended to study surface-to-bulk elemental composition by measurements with various K to understand the size of the modified region.

Ans: By conservation of energy, the kinetic energy has to be equal the change in potential energy, so KE=qV. The energy of the electron in electron-volts is numerically the same as the voltage between the plates. For example, a 5000-V potential difference produces 5000-eV electrons. Whereas, for EDS analysis 10 KV potential is used so the electrons will have kinetic energy of 10 KeV.

Here we can calculate the penetration depth of electrons in the Al surface by using the Kanaya–Okayama model (K. Kanayat 1972) { https://www.globalsino.com/EM/page4795.html}

Where, for Al targets.

       R -- Penetration Depth
       A -- Atomic Weight (g/mole)=26.98153
       n -- A constant an E0 > 5 keV so its value is 1.67
       E0 -- Beam Energy (KeV)=10 KeV
       Z -- Atomic number= 13
       ρ -- Density =2.7 g/cm3

Now replacing all these values in equation (4) we will get the value of Penetration Depth that comes out to be 1.315 µm for an electron beam in to the Al target that is surface to bulk penetration depth.

While laser beam penetration depth can be calculated from the following formula (https://en.wikipedia.org/wiki/Penetration_depth)

                                   (5)

Here penetration depth is represented by , and  is absorption co-efficient of 0.248 µm (248 nm) laser beam in Al is about 1.4711 X 106 /cm (https://refractiveindex.info/?shelf=main&book=Al&page=Rakic).

 By using this value in equation (5) we can find the penetration depth of KrF laser beam in to the Al target that is about 0.6797 µm.

Here we can see that the penetration depth of electron beam during EDS analysis is higher in Al as compared to laser beam during laser ablation experiment. Still from EDS point analysis in table given. We have seen that changes in chemical composition are still observed.

 After laser irradiation the vaporization of target takes place which causes the generation of high temperature and high pressure plasma (A. M. Mostafa 2017). This plasma during expansion introduces pressure and compressive waves in giga pascal (GPa) range that causes propagation of shock waves on and into (bulk) the irradiated target surface. Due to which the material gets plastically deformed due to enhancement in peak pressure. When it becomes higher than dynamic yield stress/strain, the generation of compressive stresses and material’s resistance to fatigue corrosion and cracking get enhanced (confirmed from EDS and XRD analysis too) (A. M. Mostafa 2017). This causes the generation of surface structures on the surface and also variation in the chemical composition on the surface as well as into the bulk material. That’s why we got variation in chemical composition in EDS results due to presence of shock effected zones into the bulk material after laser irradiation.

Elements

Untreated

In Argon

In Oxygen

Al

97

99.0

65

Ar

-

1

-

O

3

-

35

  • Some estimations of surface temperature during irradiation need to be provided.

KrF Excimer irradiated surface temperature can be calculated by using following relation (Duley 1996),

                 (5)

Here room temperature is denoted by T0, I0 is the laser beam intensity, surface reflectivity is represented by R0 (0.926 for 248 nm laser beam in Al), K is thermal diffusivity (79.5 W/mK), the absorption co-efficient is represented by  (1.4711 X 10 6 /cm) and tp is pulse duration of laser beam (1.8 X 10-8 sec) . By using these values in the above equation we can find the value of surface temperature that comes out to be 1.549 X 103 K for minimum fluence value (0.86 J.cm -2) and 2.26 X 103 K for maximum fluence value of 1.27 J.cm-2.

  • Figure 7 caption needs to be improved.

Ans: Figure captions of figure 7 that is now figure 8 is improved in the manuscript.

7) For nanohardness analysis, the measurement error has to be introduced and images of typical imprints of the indenter.

Ans: The measurement error (standard deviation) was about ± 0.03 and is also introduced in the manuscript. We have directly evaluated the data using nanohardness analysis and don’t have images of typical imprints of the indenter with us at this time. So are not in the position to include it in the manuscript.

Submission Date

28 April 2021

Date of this review

08 May 2021 20:23:02

  1. M. Mostafa, M. F. H., S. S. Obayya (2017). "Effect of laser shock peening on the hardness of AL-7075 alloy." Journal of King Saud University - Science In press: xxx-xxx.

Duley, W. W. (1996). U.V lasers effects and applications in material science. UK, Cambridge University Press

https://en.wikipedia.org/wiki/Penetration_depth.

https://refractiveindex.info/?shelf=main&book=Al&page=Rakic.

  1. Kanayat, S. O. (1972). "Penetration and energy-loss theory of electrons in solid targets." J. Phys. D: Appl. Phys. 5: 43-57.

  1. H. Iqbal, S. B., Mu. S. Rafique, A. Dawood, M. Akram, K. Mahmood, A. Hayat (2015). "Pulsed laser ablation of Germanium under vacuum and hydrogen environments at various fluences." App. Surf. Sci. 344: 146-158.

Reviewer 2 Report

The authors made a comprehensive characterization of changes induced in the aluminum sample after ablation by a nanosecond KrF excimer laser with different treatment parameters and in different environments (O2, Ar). The analysis covered the surface morphology, chemical and phase composition of the material, hardness, residual stress, etc. The course of the experiment does not raise any objections, however, the description of the results sometimes takes the form of a research report rather than a scientific study, because there is no deeper discussion of the consequences of the effects found. The work also requires a detailed editorial correction due to numerous editing errors in the text, punctuation errors, ambiguous mental shortcuts, lack of consistency in markings, freedom in the use of upper and lower case letters, etc. However, the presented comments do not negate the cognitive value of the article.

Detailed comments:

  1. Lack of information about the number of nanhardness measurements for individual variants, the graphs in Fig. 11 show that only one hardness measurement was performed for each fluence. If this is the case, the results may be very biased. There is also no information about the measurement error / standard deviation.
  2. It is necessary to provide the parameters of the X-ray analysis XRD. The description in the “Experimental Setup” chapter is limited to the name of the diffractometer only
  3. It is a pity that EDS studies were limited to a laser fluence of 0.86 Jcm − 2, in contrast to, for example, XRD studies performed for various fluences of 0.86, 1, 1.13 and 1.27 Jcm − 2. What was the reason for this?
  4. The pressure values given in the title of Table 1 and in the description are different

    Text: “Table 1 shows the variation in elemental composition (EDS data) of unexposed and exposed Al targets, exposed to 100 laser pulses, laser fluence of 0.86 Jcm−2 and at pressure of 100 torr in Ar and O2 environments”. 

    Title of the table 1: “Table 1. An EDS analysis of the unirradiated and nanosecond laser irradiated Al targets, at a laser fluence of 0.86 Jcm−2, in Ar and O2 environment at the pressure of 133 mbar.

  5. The following sentence "Whereas, X’Pert PRO (MPD) X-ray diffractometer and EDS (S3700N) were performed to analyze crystallographic structures, dislocation densities, residual strains and composition of the ablated area of the irradiated targets" shows that both the diffractometer and EDS analyzer were used to perform all the tests mentioned in the sentence, and in fact the EDS analyzer was only used to identify the chemical composition.
  6. The following sentence contains abbreviations that the Authors do not explain “Convex lens with f = 50 cm, was used to focus laser beam on the target surface at 90º w.r.t target surface”.
  7. The following sentence “All these properties make it useful in the field of process and aerospace industry as well as in engineering” contains imprecise information. What engineering do they mean? Chemical Engineering ?, surface engineering ?, mechanical engineering? It is necessary to detail.
  8. The phrase "much longer life time" in the sentence "Alumina is high performance ceramic material with excellent temperature resistance, high mechanical strength, good tribological properties, and much longer life time" is ambiguous and imprecise, because the authors do not specify in relation to what material alumina is characterized by longer life time. If they mean pure aluminum, the comparison used in this case is at least unfortunate because the "life time" of the material depends on the type of factors affecting this material.
  9. The authors presented the characteristics of the KrF laser (Abstract and Experimental Setup) in a very short and general form "KrF Excimer laser (20 ns, 248 nm, 20 Hz)". The values characterizing the laser should be given with the exact name of the parameter they refer to.
  10. The work requires a detailed editorial correction due to numerous editorial errors in the text,, punctuation errors, ambiguous mental shortcuts, lack of consistency in markings, freedom in the use of upper and lower case letters, etc.

e.g.: “KrF excimer - KrF Excimer”;

Residual Stress – Residual strain (Fig. 7, 9)”;

“………………after irradiation with 100 pulses of Excimer laser”;  

“…. characteristics of nano-second pulsed laser ablated Aluminum (Al) has been revealed”;

“….. can diffuse into the Al surface. while, during irradiation in ambient environment …..”;

„…. were utilized to investigate the Surface morphology…. „

and many others

11. Total freedom in the way of writing literature items, there are also numerous editing errors, e.g. „Ultrafastlasertexturedsiliconsolarcells”, „Laser-in- duced”, „ce- ramic”, „Quantum Electronics 37 () ()

Author Response

Dear Editor,

We have revised the manuscript according to the reviewer’s comments.

All issues raised are addressed point by point and a file dealing with answers to reviewer’s comments is attached separately.

All changes are incorporated in the manuscript also.

 All the answers to reviewer’s comments are given in RED and all changes made in the manuscript according to reviewer’s comments are also highlighted with RED color.

Regards

Nisar Ali

Comments and Suggestions for Authors

The authors made a comprehensive characterization of changes induced in the aluminum sample after ablation by a nanosecond KrF excimer laser with different treatment parameters and in different environments (O2, Ar). The analysis covered the surface morphology, chemical and phase composition of the material, hardness, residual stress, etc. The course of the experiment does not raise any objections, however, the description of the results sometimes takes the form of a research report rather than a scientific study, because there is no deeper discussion of the consequences of the effects found. The work also requires a detailed editorial correction due to numerous editing errors in the text, punctuation errors, ambiguous mental shortcuts, lack of consistency in markings, freedom in the use of upper and lower case letters, etc. However, the presented comments do not negate the cognitive value of the article.

Detailed comments:

1. Lack of information about the number of nanhardness measurements for individual variants, the graphs in Fig. 11 show that only one hardness measurement was performed for each fluence. If this is the case, the results may be very biased. There is also no information about the measurement error / standard deviation.

Ans: The nano-hardness measurements were taken from five different points on the irradiated spot and took their average. The data given in the graphs is basically showing the averaged values and the results are obviously not biased. This point is also mentioned in the manuscript. The measurement error (standard deviation) was about ± 0.03.  `

2. It is necessary to provide the parameters of the X-ray analysis XRD. The description in the “Experimental Setup” chapter is limited to the name of the diffractometer only.

Ans:  Corrections have made and highlighted in the manuscript.

 X’Pert PRO-MPD X-ray diffractometer working in θ–θ mode, with 40 mA current and 40KV voltage was used for crystallographic, dislocation density, residual stress measurements and for study of variation in composition of laser ablated Al targets.

3. It is a pity that EDS studies were limited to a laser fluence of 0.86 Jcm − 2, in contrast to, for example, XRD studies performed for various fluences of 0.86, 1, 1.13 and 1.27 Jcm − 2. What was the reason for this?

Ans: we have chosen EDS just to support the XRD and Raman results, as this technique is not available at our institution and is a paid technique that’s why we have done it only for three samples. We studied the chemical composition of unirradiated and laser irradiated Al targets in Ar and O2 ambient and gave their comparison.

4. The pressure values given in the title of Table 1 and in the description are different

Text: “Table 1 shows the variation in elemental composition (EDS data) of unexposed and exposed Al targets, exposed to 100 laser pulses, laser fluence of 0.86 Jcm−2 and at pressure of 100 torr in Ar and O2 environments”. 

Title of the table 1: “Table 1. An EDS analysis of the unirradiated and nanosecond laser irradiated Al targets, at a laser fluence of 0.86 Jcm−2, in Ar and O2 environment at the pressure of 133 mbar.

Ans: correction is made and highlighted in the manuscript.

5. The following sentence "Whereas, X’Pert PRO (MPD) X-ray diffractometer and EDS (S3700N) were performed to analyze crystallographic structures, dislocation densities, residual strains and composition of the ablated area of the irradiated targets" shows that both the diffractometer and EDS analyzer were used to perform all the tests mentioned in the sentence, and in fact the EDS analyzer was only used to identify the chemical composition.

Ans: The sentence is corrected as per guidance of reviewer and replaced by “X’Pert PRO-MPD X-ray diffractometer working in mode of θ–θ, with 40 mA current and 40Kv voltage was used for crystallographic, dislocation density, residual stress measurements and for study of variation in composition of laser ablated Al targets. While the EDS (S3700N) analyzer was used for chemical compositional analysis.

6. The following sentence contains abbreviations that the Authors do not explain “Convex lens with f = 50 cm, was used to focus laser beam on the target surface at 90º w.r.t target surface”.

Ans: The sentence is corrected and highlighted in the manuscript accordingly and replaced by “Convex lens with 50 cm focal length was used to focus laser beam on the target surface at 90º with respect to target surface.”

7. The following sentence “All these properties make it useful in the field of process and aerospace industry as well as in engineering” contains imprecise information. What engineering do they mean? Chemical Engineering ?, surface engineering ?, mechanical engineering? It is necessary to detail.

Ans: The phrase “All these properties make it useful in the field of process and aerospace industry as well as in engineering” Is replaced by

“All these properties make it useful in the field of process and aerospace industry as well as in surface engineering”

In the manuscript.

8. The phrase "much longer life time" in the sentence "Alumina is high performance ceramic material with excellent temperature resistance, high mechanical strength, good tribological properties, and much longer life time" is ambiguous and imprecise, because the authors do not specify in relation to what material alumina is characterized by longer life time. If they mean pure aluminum, the comparison used in this case is at least unfortunate because the "life time" of the material depends on the type of factors affecting this material.

Ans: This ambiguity is removed and highlighted from the manuscript and is replaced by,

“Alumina is high performance ceramic material with excellent temperature resistance, high mechanical strength, good tribological properties”.

9. The authors presented the characteristics of the KrF laser (Abstract and Experimental Setup) in a very short and general form "KrF Excimer laser (20 ns, 248 nm, 20 Hz)". The values characterizing the laser should be given with the exact name of the parameter they refer to.

Ans: The abbreviations are replaced by exact names of parameters used in abstract as well as in experimental setup.

10. The work requires a detailed editorial correction due to numerous editorial errors in the text,, punctuation errors, ambiguous mental shortcuts, lack of consistency in markings, freedom in the use of upper and lower case letters, etc.

e.g.: “KrF excimer KrF Excimer”;

Residual Stress – Residual strain (Fig. 7, 9)”;

“………………after irradiation with 100 pulses of Excimer laser”;  

“…. characteristics of nano-second pulsed laser ablated Aluminum (Al) has been revealed”;

“….. can diffuse into the Al surface. while, during irradiation in ambient environment …..”;

„…. were utilized to investigate the Surface morphology…. „

11. Total freedom in the way of writing literature items, there are also numerous editing errors, e.g. „Ultrafastlasertexturedsiliconsolarcells”, „Laser-in- duced”, „ce- ramic”, „Quantum Electronics 37 () ()

Ans: The manuscript is read carefully and corrections are being made in the manuscript.

Submission Date

28 April 2021

Date of this review

09 May 2021 11:45:21

Round 2

Reviewer 1 Report

The authors have addressed all comments. However, the presentation of results still needs to be improved both in graph quality and presentation clarity. Probably, dome images and part of the text can be moved to supporting material, if possible.

Author Response

Dear Editor

We are submitting the revised manuscript entitled

Study of micro/nano structuring and mechanical properties of KrF excimer laser irradiated Al for aerospace industry and surface engineering applications

Authors: Umm-i-Kalsoom, Nisar Ali, Shazia Bashir, Ali Mohammad Alshehri, Narjis Begum

for consideration of publication in Materials.

Please acknowledge it.

With kind regards

Nisar Ali

Respected Reviewer

We have revised the manuscript according to the reviewer’s comments.

All issues raised are addressed and a file dealing with answers to reviewer’s comments is attached separately.

All changes are incorporated in the manuscript also.

 All the answers to reviewer’s comments are given in RED and all changes made in the manuscript according to reviewer’s comments are also highlighted with RED color.

Regards

Nisar Ali

Open Review

English language and style

( ) Extensive editing of English language and style required
( ) Moderate English changes required
( ) English language and style are fine/minor spell check required
(x) I don't feel qualified to judge about the English language and style

Yes

Can be improved

Must be improved

Not applicable

Does the introduction provide sufficient background and include all relevant references?

( )

(x)

( )

( )

Is the research design appropriate?

(x)

( )

( )

( )

Are the methods adequately described?

(x)

( )

( )

( )

Are the results clearly presented?

( )

( )

(x)

( )

Are the conclusions supported by the results?

(x)

( )

( )

( )

Comments and Suggestions for Authors

Q:The authors have addressed all comments. However, the presentation of results still needs to be improved both in graph quality and presentation clarity. Probably, dome images and part of the text can be moved to supporting material, if possible.

Ans: The results are improved both in graph quality and presentation clarity also the relevant references are induced in the introduction part to enhance its clarity.
